# BayesGaze: A Bayesian Approach to Eye-Gaze Based Target Selection

Zhi Li*
Stony Brook University

Maozheng Zhao†
Stony Brook University

Yifan Wang ‡
Stony Brook University

Sina Rashidian§
Stony Brook University

Furqan Baig¶
Stony Brook University

Rui Liu‖
Stony Brook University

Wanyu Liu**
IRCAM Centre Pompidou

Michel Beaudouin-Lafon ††
Université Paris-Saclay

Brooke Ellison‡‡
Stony Brook University

Fusheng Wang
Stony Brook University

IV Ramakrishnan
Stony Brook University

Xiaojun Bi
Stony Brook University

## ABSTRACT

Selecting targets accurately and quickly with eye-gaze input remains an open research question. In this paper, we introduce BayesGaze, a Bayesian approach of determining the selected target given an eye-gaze trajectory. This approach views each sampling point in an eye-gaze trajectory as a signal for selecting a target. It then uses the Bayes' theorem to calculate the posterior probability of selecting a target given a sampling point, and accumulates the posterior probabilities weighted by sampling interval to determine the selected target. The selection results are fed back to update the prior distribution of targets, which is modeled by a categorical distribution. Our investigation shows that BayesGaze improves target selection accuracy and speed over a dwell-based selection method, and the Center of Gravity Mapping (CM) method. Our research shows that both accumulating posterior and incorporating the prior are effective in improving the performance of eye-gaze based target selection.

**Index Terms:** Human-centered computing—Human computer interaction (HCI); Human-centered computing—Human computer interaction (HCI)—Interaction techniques; Human-centered computing—Human computer interaction (HCI)—HCI design and evaluation methods—User studies;

## 1 INTRODUCTION

Selecting a target with the gaze remains a central problem of eye-based interaction. Two factors make this problem challenging [18]. First, gaze input is noisy because of both inadvertent eye movements and inevitable noise in the tracking device [50]. Therefore, it is difficult for a user to move their gaze to a particular position and stabilize it for an extended period of time. Second, unlike using a mouse, where a user can confirm the selection by clicking a button, gaze-based interaction lacks an easy-to-use approach to confirm the selection, adding a layer of difficulty to the design of a selection technique [43]. Although previous research has explored target selection using dwell [15, 17], motion correlation [45] and dynamic user interfaces [26, 30, 40], quickly and accurately selecting a target with gaze input remains an open research question.

*zhili3@cs.stonybrook.edu
†mazhao@cs.stonybrook.edu
‡yifanwang@cs.stonybrook.edu
§srashidian@cs.stonybrook.edu
¶fbaig@cs.stonybrook.edu
‖ruiliu1@cs.stonybrook.edu
**AbbyWanyu.Liu@ircam.fr
††mbl@lri.fr
‡‡brooke.ellison@stonybrook.edu
  fushwang@cs.stonybrook.edu
  ram@cs.stonybrook.edu
  xiaojun@cs.stonybrook.edu

Inspired by the literature showing that Bayes' theorem is a promising principle for handling uncertainty and noise in input signals (e.g., [4, 52]), we investigate how to apply a Bayesian perspective to determining the selected target given a gaze trajectory. Applying Bayes' theorem to gaze-based target selection raises two main challenges. First, it is not clear how to obtain the likelihood function for a gaze trajectory that contains a sequence of input signals (gaze points), i.e. the probability of observing a gaze trajectory given the target. Second, unlike touch or mouse input, which have a clear definitions of the terminal moment of the input, e.g. lifting the finger from the touch screen or mouse button, gaze input lacks a clear delimiter of the completion of a selection action. It is therefore necessary to design a method to determine when the selection action is completed.

To address these challenges, we introduce BayesGaze (Figure 1), a Bayesian approach for determining the selected target given a gaze trajectory. This approach first views each sampling point in a gaze trajectory as a signal for selecting a target, and then uses Bayes' theorem to calculate the posterior probability of selecting a target given a sampling point. The likelihood of a target being selected is based on the distance between the sampling point and the target center, and the prior probability of a target being selected is modeled by a categorical distribution and updated after a selection action. BayesGaze then accumulates the posterior probabilities over all sampling points, weighted by the sampling interval, to determine the selected target. BayesGaze advances the Center of Gravity Mapping (CM) [4] by modeling the prior and incorporating it into the process of determining the selected target. This contribution is key to improving the performance of gaze-based target selection.

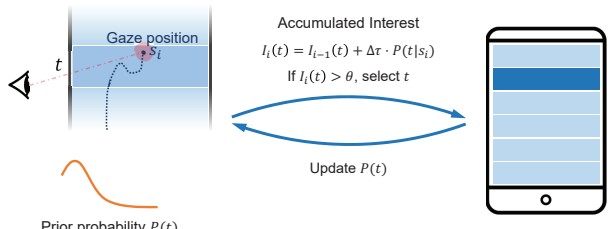

Figure 1: An overview of how BayesGaze works. Given a gaze position $s_i$ sampled at time $i$ in a gaze trajectory, BayesGaze updates the accumulated interest of selecting target $t$, denoted by $I_i(t)$, by adding $P(t|s_i)$ weighted by the sampling interval $\Delta\tau$ to $I_{i-1}(t)$. $P(t|s_i)$ is the posterior probability of selecting $t$ given $s_i$, which is calculated based on Bayes' theorem. If the accumulated interest $I_i(t)$ exceeds a threshold $\theta$, the target $t$ is selected. BayesGaze then updates the prior probability $P(t)$ accordingly.

We report on a controlled experiment showing that BayesGaze improves target selection accuracy (from 82.1% to 88.3%) and speed (from 2.49 seconds per selection to 2.23 seconds) over a dwell-based selection method. BayesGaze also outperforms the CM method [4].

Overall, our investigation shows that accumulating the posterior probability and incorporating the prior are effective in improving the performance of gaze-based target selection.

## 2 RELATED WORK

BayesGaze builds on previous work on gaze input and on Bayesian approaches. Here we review related work in gaze-based target selection techniques, Bayesian approaches to gaze input, and gaze-tracking technology.

### 2.1 Gaze Based Target Selection

Gaze-based target selection is a key technique for supporting a number of gaze interaction technologies such as gaze-based text input [36], gaming [16] or smart device control [37]. Dwell-based target selection (Dwell) [15, 17, 51] is the most well-known and most widely used target selection method. It requires a user to dwell their gaze on a target for a specific uninterrupted period of time (usually several hundred milliseconds to 1 or 2 seconds) to select it. Such a highly concentrated action often results in eye fatigue [34]. Many works have been devoted to improving the Dwell technique by enabling a shorter dwell time and to finding other gaze-based target selection methods. For example, letting a user adjust the dwell time manually can lead to a shorter dwell time, from 876 ms to 282 ms [27]. Previous research [15] used Fitts' law to model gaze input and suggested selecting the target once the user's gaze fixates the target. Other works have explored adjusting the dwell time based on how likely the target will be selected [32, 35].

In addition to dwell-based methods, researchers have proposed alternatives to improve gaze-based target selection from the two perspectives: handling the noisy gaze input and designing new selection action [18, 50]. To accommodate the inaccuracy of eye-gaze input, some works used dynamic expansion/zooming of the display [30,40] or new UIs, e.g. Actigaze [26] used a set of confirmation buttons to make gaze target selection easier. Other works investigated error-aware gaze target selection so that the inaccuracy of target selection can be tracked and the system can provide design guidelines for UIs [3, 11]. Gaze target selection actions are also well explored. For example, *motion correlation* between the target movement and gaze trajectory has been proposed to determine the selected target [45]. Actions such as blinking [7] and gaze gesture [9] have also been explored for target selection. Previous research has also used multimodal input to get rid of the dwelling action. For example, once the user gazes at the target, a separate input, such as a keyboard input [22], hand-held touchscreen input [43], EMG input [29] and the head movement [39] can be employed to perform the selection action. However, incorporating other input modalities require more effort from users and extra input device, and it may be infeasible for some users, such as amyotrophic lateral sclerosis (ALS) patients with motor disability, who are not able to use a keyboard or a hand-held touchscreen, neither move their heads.

### 2.2 Bayesian Approaches to Target Selection

There is a growing interest in applying a Bayesian perspective to handle uncertainty in target selection. Some of this research is related to gaze input. For example, previous research has proposed probabilistic frameworks to deal with uncertainty in the input process, such as handling the uncertainty of touch actions on mobile devices [6,46] and touchscreens [52], and also handling uncertainty in gaze-based interactions [4,33].

Our work is related to the recent work BayesianCommand, which uses Bayes' theorem to handle uncertainty in touch target selection and word-gesture input [52]. The fundamental difference between our work and BayesianCommand is that in our work, gaze input does not have well-defined starting or ending moments, but touch input does (i.e., landing a finger on screen to start input, and taking finger off to end the input). Therefore, BayesianCommand cannot be applied to gaze input directly.

Our research is also related to previous work on using a Bayesian perspective to address the gaze-to-object mapping problem, i.e. the *Center of Gravity Mapping method (CM)* [4]. CM is an improved version of the FM algorithm [48], which performed the best among 9 extant gaze-to-object mapping algorithms [41]. The main difference between our work and CM is that CM does not model nor update the prior, while our approach incorporates the prior into the process of deciding the selected target, which turns out to be the primary reason why BayesGaze improves target selection accuracy and reduces selection time. Furthermore, BayesGaze is designed for the gaze target selection problem while the CM is designed for gaze-to-object mapping problem. The gaze-based target selection is a different problem than gaze-to-object mapping [4, 41] because the former requires a mechanism to commit the selection while the latter does not.

### 2.3 Gaze Tracking Technology

Gaze tracking technology is becoming increasingly mature and available. For example, a number of professional gaze trackers are available, including Tobii 4C [23], SMI REDn [20] or Eyelink 1000 plus [38], that cost several hundreds up to a few thousand dollars. Previous research has also enabled gaze tracking with off-the-shelf cameras by using a fisheye camera [2], the front-facing RGB camera of a tablet [47], or by leveraging the glint of the screen on the user's cornea [14]. Deep learning techniques have also been used to predict gaze position using Convolutional Neural Networks [19,49].

Unlike the above approaches, we enabled gaze tracking with an off-the-shelf and widely used iPad Pro using a true depth camera and powered by Apple's ARKit, with which eye gaze can be measured directly without an extra gaze tracking device.

## 3 BAYESGAZE: A BAYESIAN PERSPECTIVE ON GAZE TARGET SELECTION

### 3.1 A Formal Description of the Gaze Based Target Selection Problem

The gaze-based target selection problem can be formally described as the following research question. Given a gaze trajectory, which one is the intended target among a set of candidates denoted by $\mathbb{T} = \{t_1, t_2, \ldots, t_N\}$?

As shown in previous research [4, 41, 48], the existing algorithms for solving the gaze-based target selection problem can be described through an interest accumulation framework: each target candidate (denoted by $t$) accumulates a certain amount of "time" or "interest" from gaze input, until one of them reaches a threshold (denoted by $\theta$) for being selected. Under this framework, the widely adopted dwell-based target selection method can be expressed as follow.

*Dwell-based Target Selection Method.* Assuming that the gaze trajectory is denoted by $\mathbb{S} = \{s_1, s_2, \ldots, s_K\}$ where $s_i$ is a sampling point along the gaze trajectory at time $i$, the accumulated "interest" for a target candidate $t$ at time $i$, denoted by $I_i(t)$, is calculated as:

$$I_i(t) = \begin{cases} I_{i-1}(t) + \Delta\tau, & \text{if } s_i \text{ is within the target } t \\ 0, & \text{otherwise} \end{cases} \quad (1)$$

where $s_i$ is the gaze position at time $i$, and $\Delta\tau$ is the sampling interval. $I_i(t)$ represents the duration during which the gaze position stayed continuously within the target candidate $t$. If the gaze position moves outside the target, it resets $I_i(t)$ to 0. To select a target, the eye-gaze position needs to continuously stay within a target for a period of $\theta$. In other words, the selected target is the one (denoted by $t^*$) whose accumulated selection interest $I_i(t^*)$ first reaches $\theta$ (i.e., $I_i(t^*) \geq \theta$).

## 3.2 The BayesGaze Algorithm

Under the framework of "accumulating selection interest", we propose BayesGaze, a Bayesian perspective for gaze-based target selection. It views each sampling point in a gaze trajectory as a signal for selecting a target, and then uses Bayes' theorem to calculate the posterior probability of selecting a target given a sampling point. BayesGaze then accumulates the posterior probabilities over all sampling points weighted by the sampling interval, as accumulated interest of selecting a target. A target candidate will be selected once the accumulated interest reaches a threshold $\theta$. Formally, the accumulated interest of selecting a target $t$ is calculated as follows, given the sampling point $s_i$:

$$I_i(t) = I_{i-1}(t) + \Delta\tau \cdot P(t|s_i). \tag{2}$$

The posterior $P(t|s_i)$ can be estimated according to Bayes' theorem, assuming there are $N$ target candidates:

$$P(t|s_i) = \frac{P(s_i|t)P(t)}{P(s_i)} = \frac{P(s_i|t)P(t)}{\sum_{j=1}^{N} P(s_i|t_j)P(t_j)}, \tag{3}$$

where $P(t)$ is the prior probability of target $t$ being the intended target without observing the current gaze input trajectory, and $P(s_i|t)$ is the probability of $s_i$ if the intended target is $t$ (the likelihood).

BayesGaze has the following characteristics. First, BayesGaze resumes the accumulation of selection interest from where it left if the gaze trajectory accidentally leaves a target but returns to it later. It address a problem of dwell-based method (Equation 1) that if the eye-gaze position moves outside a target, the accumulated interest for selecting such a target is reset to 0. Second, it weights the accumulated interest with the distance between the gaze point and the target center, through the likelihood function $P(s_i|t)$. The closer a gaze point is to the target center, the more "interest" such a point will contribute to the target selection. Third, it updates the prior distribution of targets ($P(t)$) and incorporate it into the procedure of deciding the selected target.

In the following part, we introduce how to estimate the prior distribution $P(t)$ and the likelihood $P(s|t)$, which are keys for applying BayesGaze.

### 3.2.1 Prior Probability Model

This part introduces a frequency model to estimate the prior distribution $P(t)$ based on the observable target selection history. We assume that the user does not select targets randomly and the target selection follows some distribution, e.g. Zipf's Law. This assumption is made based on the selection patterns in menu selection [8, 25, 52], smartphone APP launching [31], and command triggering [1, 10, 52]. All of them are tasks that gaze target selection can support.

We model the prior distribution (i.e., a target candidate being selected prior to observing the current gaze trajectory) as a categorical distribution. More specifically, the outcome of a gaze-based selection trial that results in a selected target is viewed as a random variable $x$ whose value is one of $N$ categories (the $N$ target candidates). The core parameter of this random variable $x$ is the parameter vector $\boldsymbol{p} = (P(t_1), P(t_2), ..., P(t_N))$, which describes the probability of each category. As a common practice in Bayesian inference, we also view this parameter vector $\boldsymbol{p}$ as a random variable and give it a prior distribution, using the Dirichlet distribution.

According to the properties of Dirichlet distributions, after each target selection trial we can update the expected value of the posterior $\boldsymbol{p}$ as follows:

$$P(t_i) = \frac{k + c_i}{k \cdot N + \sum_{j=1}^{N} c_j}, \tag{4}$$

where $N$ is the number of candidate targets (e.g., the number of menu items), $c_i$ is the number of times we have observed target $t_i$

being selected, and $k$ is the pseudocount of the Dirichlet prior, a hyper-parameter of the distribution. The parameter $k$ can also be viewed as the update rate, which is a positive constant that controls how quickly the $P(t_i)$ are updated. Note that the prior updating model (Equation 4) is the same as the model proposed by Zhu et al. [52], although these authors do not describe it under the paradigm of categorical-Dirichlet distributions. We use the expected value of $\boldsymbol{p}$ (Equation 4) as the prior model in BayesGaze (Equation 3).

This prior model matches our expectations well. When there is no target selection observed, the probability $P(t_i)$ is $\frac{k}{k \cdot N} = \frac{1}{N}$, which means that all candidate targets have equal probability. Whereas when there are enough target selections observed, i.e. $c_i \gg k$, we have $P(t_i) \approx \frac{c_i}{\sum_j c_j}$, which means that $P(t_i)$ can be estimated based on the frequency of $t_i$ having been selected before.

By setting different $k$, we can balance $P(t_i)$ between two extreme cases: 1) when $k \to +\infty$, we have $P(t_i) \approx \frac{1}{N}$, that is, the prior probabilities of all candidate targets are the equal. 2) when $k = 0$, we have $P(t_i) = \frac{c_i}{\sum_j c_j}$, which means that the prior probability is only based on the history selection frequency. We later use empirical data to determine an optimal value for $k$.

### 3.2.2 Likelihood Model

The goal of this step is to estimate $P(s_i|t)$, the likelihood of observing $s_i$ if $t$ is the intended target. Since $s_i$ is a single gaze position, a reasonable assumption is that $P(s_i|t)$ is higher if $s_i$ is closer to the center of $t$. We follow Bernard et al. [4] and use a Gaussian density function to describe the likelihood of observing $s_i$, a common method for modeling likelihood for a single-point target selection:

$$P(s_i|t) = \frac{1}{\sqrt{2\pi\sigma^2}} \exp(-\frac{||s_i - c_t||^2}{2\sigma^2}), \tag{5}$$

where $c_t$ is the center of target $t$, the term $||s_i - c_t||$ is the $L^2$ Euclidean norm of the vector $s_i - c_t$, $\sigma$ is an empirical parameter defining how concentrated should the gaze points be. The parameter $\sigma$ controls how much interest can be accumulated at a certain distance. If $\sigma$ is too small, a target accumulates high interest only when the gaze point is close to the target center, which could make the target hard to select. On the other hand, if $\sigma$ is too large, the accumulated interests for neighboring targets could become large and cause mis-selections. We estimate an optimal $\sigma$ from real data in the next section.

---

**Algorithm 1** BayesGaze Algorithm

---

**Input:** Target set: $\mathbb{T} = \{t_1, t_2, \ldots, t_N\}$, Gaze trajectory: $\mathbb{S} = \{s_1, s_2, \ldots, s_K\}$, Threshold: $\theta$
**Output:** Selected target $t$, Selection time: $\tau_{sel}$
1: **for** $s_i$ **in** $S$ **do**
2:     **for** $t_j$ **in** $\mathbb{T}$ **do**
3:         Obtain prior probability $P(t_j)$ and compute likelihood $P(s_i|t_j)$ using Equation 5;
4:         Compute accumulated interest $I_i(t_j)$ from Equation 2;
5:         **if** $I_i(t_j) > \theta$ **then**
6:             Update prior probability $P(t_m)$ for each $t_m \in \mathbb{T}$ given that $t_j$ is selected using Equation 4;
7:             **return** $t_j, i \cdot \Delta\tau$
8:         **end if**
9:     **end for**
10: **end for**

---

After obtaining both the prior probability and the likelihood, we can use BayesGaze to perform target selection. The BayesGaze algorithm is summarized in Algorithm 1. Note that the algorithm can be run online, i.e. when a gaze point $s_i$ is sampled by the gaze

tracker, the top-level for-loop can be executed to check if a target is selected.

### 3.2.3 BayesGaze without Prior

If we consider the prior to be Uniform distribution before every trial (i.e. $\forall t_i \in \mathbb{T}, P(t_i) = 1/N$), BayesGaze will be identical to the *Center of Gravity Mapping (CM)* algorithm [4] (referred to as the CM method hereafter), a previously proposed method for deciding a target for a gaze-to-object mapping task. Under this special condition, the accumulated interest of the CM method can be calculated by Equation 2 with the prior $P(t) = 1/N$, that is:

$$I_i(t) = I_{i-1}(t) + \Delta\tau \cdot P(t|s_i) = I_{i-1}(t) + \Delta\tau \cdot \frac{P(s_i|t)}{\sum_{j=1}^{N} P(s_i|t_j)}, \quad (6)$$

where $P(s_i|t)$ is calculated by Equation 5. Therefore, we view BayesGaze as an improvement over the CM method that updates and incorporates the prior in the target selection process. The CM method is also very similar to the previously proposed Fractional Mapping method [41, 48]. We later compare BayesGaze with the CM method to examine to what degree incorporating the prior can improve gaze target selection performance.

In order to successfully apply the BayesGaze algorithm, we need to obtain the values of three parameters, denoted as a 3-tuple $[k, \sigma, \theta]$, where $k$ is part of the prior probability model (Equation 4), $\sigma$ is part of the likelihood model (Equation 5), and $\theta$ is the threshold of the accumulated interest for committing a selection. We carried out a study to collect gaze data for target selection and determine the optimal parameter values from that data.

## 4 PARAMETER DETERMINATION

We adopted a data-driven simulation approach to search for the optimal parameter values for the BayesGaze algorithm. The procedure consists of two phases. In Phase 1, we carried out a user study to collect gaze input data for selecting a target. In Phase 2, we fed the collected data to the BayesGaze algorithm to search for the optimal parameter values. We also searched for the optimal parameter for the Dwell method (Equation 1) and for the CM method (Equation 6).

### 4.1 Phase 1: Collecting Gaze Input Data

We first carried out a user study to collect gaze input data for selecting a target. We focused on a 1-dimensional target selection task, where the target is a horizontal bar and gaze motion is vertical. We picked this task because 1-dimensional pointing is a typical target selection task, and horizontal bars are widely used UI elements on mobile computing devices such as smartphones and tablets.

#### 4.1.1 Participants

Twelve users (4 female) between 23 and 31 years old (average 27.25±2.22) participated in the experiment. All of them had normal or correct-to-normal sight and none of them was color blind. None of them had the experience of using gaze tracking devices or applications.

#### 4.1.2 Apparatus

We used an 11-inch iPad Pro for gaze tracking and running the experiment. The gaze tracking was implemented using Apple's ARKit library, and the sampling rate was 60Hz supported by the library. Specifically, we used the *leftEyeTransform* and *rightEyeTransform* provided by ARKit library and performed a *hitTestWithSegment* call to obtain the raw gaze position. Based on the recommendation of [11], we used the Outlier Correction filter with a triangle kernel [21] to obtain smooth gaze tracking. The filter contains a saccade/fixation detection module so that it can apply sliding windows of different lengths separately for saccades and fixations. The thresholds for the $x$ and $y$ axis to detect a saccade were both set

to $0.5°$ (calculated based on the estimated face-screen distance). For fixations, the sliding window size of the filter was set to 40 as suggested by [11]. For the saccade, the sliding window size of the filter was set to 10, rather than using the raw position directly, to increase gaze tracking stability. We also followed the findings of previous works [24, 39, 42] that allow head movements to improve target selection performance. We used a gazing task where the user gazes at 40 different points on the screen with a cursor showing where the user is looking at to test the gazing accuracy. The result showed a $0.67°$ with a standard deviation of 0.85, which means the user may accurately control the gaze to select targets.

#### 4.1.3 Procedure

During the experiment the participant sat in front of a desk where an iPad Pro running the experiment was placed on a phone holder. The participant can freely adjust the iPad position, and was instructed to keep the distance between their eyes and the iPad at around 40 cm.

The study includes multiple target selection trials. In each trial, a horizontal bar in blue was displayed on the screen as the target and the participant was instructed to select it via gaze input. Figure 2 shows the setup. Before each trial, the participant first moved the gaze-controlled cursor in the starting gray bar. After 3 seconds, the starting bar turned green, signaling the start of the trial. The participant was then instructed to move the cursor with their gaze to select a target of width ($W$) at a distance $D$ from the starting bar. We collected gaze input data for 5 seconds after a trial started. We assumed that 5 seconds was long enough for the participants to select a target. After 5 seconds, a new trial starts. Each participant took a break after 15 trials. In total the experiment lasted around 15 minutes per participant.

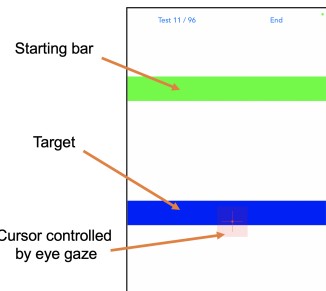

(a) A screenshot of the experiment

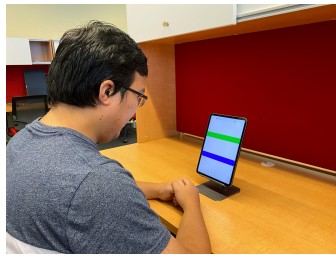

(b) A user is doing the experiment

Figure 2: A screenshot of the study. The green button is the starting bar, and the target is shown as a blue bar. There is a red cursor indicating where the participant is looking.

We adopted a within-participant $3 \times 4 \times 2$ design with three levels of target width $W$: 2 cm ($2.86°$ calculated based on a participant-screen distance of 40 cm), 3 cm ($4.29°$), and 4 cm ($5.76°$), four levels of distance $D$: 6 cm ($8.53°$), 8 cm ($11.31°$), 10 cm ($14.04°$), and 12 cm ($16.70°$), and two levels of gaze motion direction: up

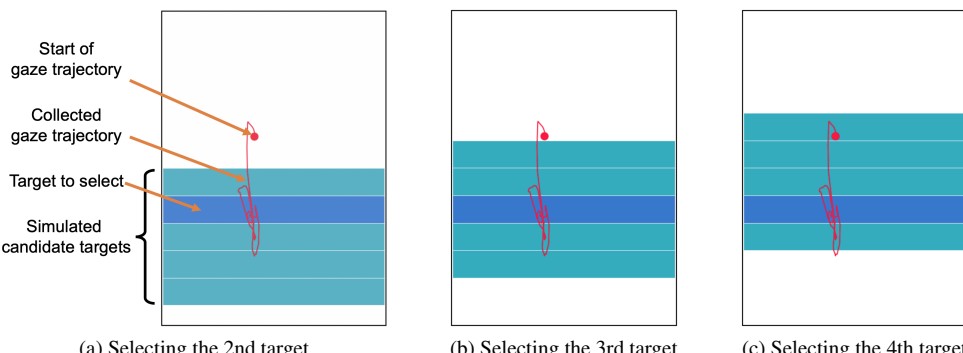

| Start of gaze trajectory |
| Collected gaze trajectory |
| Target to select |
| Simulated candidate targets |

(a) Selecting the 2nd target      (b) Selecting the 3rd target      (c) Selecting the 4th target

Figure 3: An example of using the same gaze trajectory to simulate selecting a target (the blue one) at different indices among the five horizontal bars. The three red line shows the same gaze trajectory collected in the experiment. The red dot indicates the start of the trajectory. A simulated user is selecting the 2nd (a), the 3rd (b), and the 4th (c) target among 5 target candidates, with the same gaze trajectory.

or down from the starting bar. We counterbalanced the factors by randomizing the trials in the experiment.

In total, the study resulted in 12 participant × 3 target sizes × 4 distances × 2 directions × 2 repetitions = 576 trials.

## 4.2 Phase 2: Determining Parameters from the Collected Data

We created a set of gaze-based target selection tasks, simulated gaze input based on the data collected in Phase 1, and searched for the parameter values for the BayesGaze, CM, and Dwell that led to high input accuracy and fast input speed.

### 4.2.1 Simulating Eye-Gaze Target Selection Tasks

We first created a set of target selection tasks in which a user is supposed to control their gaze to select a target among $N$ candidates. These $N$ candidates are stacked together with no gap between them to simulate the common vertical list or vertical menu design of mobile devices (e.g., settings menus in iOS). We included the same 3 target sizes in the simulation as in the data collection study (2, 3, and 4 cm) and set $N = 5$. The gaze trajectories for selecting a target are obtained from the collected data, according to the target sizes. Figure 3 shows examples of simulated gaze trajectories for selecting different targets on the screen.

Since previous research has shown that the distribution of menu items being selected follows Zipf's distribution [1, 8, 10, 25, 31, 52], we assumed that the frequency of each candidate being the target follows Zipf's Law:

$$f(l; \alpha, N) = \frac{1/l^\alpha}{\sum_{n=1}^{N} (1/n^\alpha)}, \tag{7}$$

where $N$ is the number of candidate targets (in the simulation, $N = 5$), $l \in \{1, 2, \ldots, N\}$, $n$ is the rank of each target, and $\alpha$ is the value of the exponent characterizing the distribution. We include 4 $\alpha$ values (0.5, 1, 2, 3) in the simulation.

For each target size, we had 192 collected trajectories. Among the $N$ candidates, we randomly assigned the frequencies. For example, when $N = 5$ and $\alpha = 1$, the generated frequencies can be [28, 84, 21, 42, 17], which means that the first target among 5 candidates will be selected 28 times, the second 84 times, etc. We randomly selected trajectories (without repetition) to simulate selecting targets at different indices given the generated frequencies.

### 4.2.2 Searching for the Parameter Values

Given a particular parameter tuple $[k, \sigma, \theta]$, we ran the BayesGaze algorithm to determine the selected target in the simulated target

selection tasks. We viewed the process of searching for the optimal parameter values as an optimization problem: determining parameter values that optimizes target selection performance, measured in terms of success rate and selection time.

We performed a grid search to search for optimal parameter values for $k$, $\sigma$ and $\theta$. In the grid search, $k$ ranges from 0.5 to 5 by steps of 0.5, $\sigma$ ranges from 0.14 cm (0.2°) to 1.4 cm (2°) by steps of 0.14 cm, $\theta$ ranges from 0.2 seconds to 2 seconds by steps of 0.1 seconds. The simulation results showed that different values for $k$ do not influence performance. We chose $k^*=1$, as in [52]. When $k=1$, the Dirichlet prior of the Categorical distribution, without observing any selection results, becomes a Uniform distribution, i.e. an equally distributed prior. The best parameters for $\sigma$ were from 0.28 cm to 0.56 cm for BayesGaze. We chose $\sigma^*=0.28$ cm to reduce the chance of mis-selections.

Because we want to improve two objectives, success rate and selection time, we adopted a Pareto optimization process to find the optimal $\theta$. The process generates a set of parameter values, called the Pareto-optimal set or Pareto front. Each parameter in the set is Pareto-optimal, which means that none of the two metrics (success rate or selection time) can be improved without hurting the other metric. We plot the Pareto front of BayesGaze in Figure 4a. We followed the exact same optimization process to search for the optimal parameter values for the CM and Dwell methods, and generated the corresponding Pareto fronts in Figure 4b and 4c. For the CM method, the parameters are a 2-tuple $[\sigma, \theta]$, as it does not incorporate the prior into the accumulated interest. For the Dwell method, the parameter is $\theta$, the threshold for deciding whether a target is selected based on the accumulated selection interest.

To balance the success rate and selection time, we assigned equal weights to success rate and selection time. We first normalized the success rate and selection time to the range $[0,1]$. We picked a parameter value $\theta^*$ that leads to the best overall score $S$, which is defined as:

$$S = 0.5 \times SuccessRate - 0.5 \times SelectionTime, \tag{8}$$

where $SuccessRate$ and $SelectionTime$ are the normalized values between 0 and 1, according to the highest and lowest values displayed in Figure 4. The coefficient of $SelectionTime$ is -0.5 because the lower the selection time, the higher the selection performance. The optimal parameters for different $\alpha$ values are the same and are summarized in Table 1.

## 5 A TARGET SELECTION EXPERIMENT

To empirically evaluate BayesGaze, we conducted an 1D gaze-based target selection study using the parameters from the simulations. We

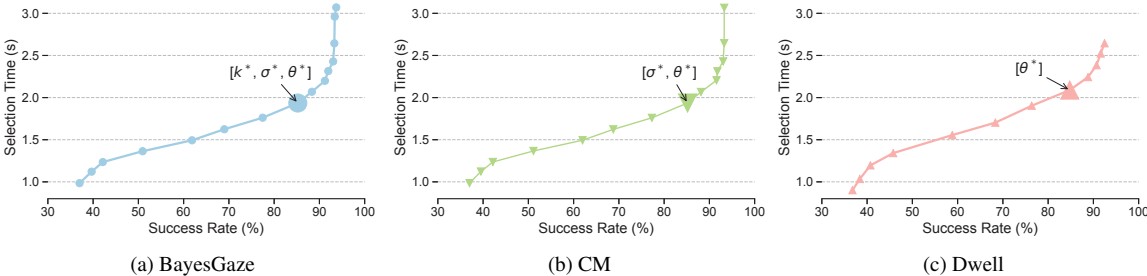

(a) BayesGaze      (b) CM      (c) Dwell

Figure 4: The Pareto front of different parameter combinations for 3 target selection methods under $\alpha = 1$ in Zipf's Law. The enlarged dots represent the selected parameter settings for three methods, respectively. These settings have the most balanced performance according to Equation 8.

| Target Selection Method | $k^*$ | $\sigma^*$ | $\theta^*$ |
|---|---|---|---|
| BayesGaze | 1 | 0.28 cm | 0.9 |
| CM | – | 0.28 cm | 0.9 |
| Dwell | – | – | 0.8 |

Table 1: Optimal parameters (same for different $\alpha$ in Zipf's Law) selected on the Pareto front for three target selection methods

included CM and Dwell as baselines in our study because (1) Dwell was a widely adopted target selection method and CM was one of the best-performed algorithms from the literature, and (2) CM can be viewed as BayesGaze without prior. Including these two methods in comparison allowed us to evaluate whether BayesGaze improved the performance with extant algorithms, and to understand how the two components of BayesGaze (likelihood function and prior) would contribute the target selection performance improvement.

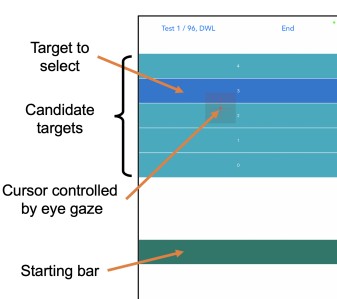

(a) A screenshot of the experiment

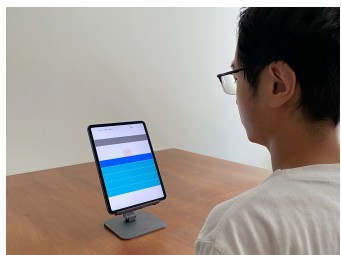

(b) A user is doing the experiment

Figure 5: The controlled 1D gaze target selection experiment

## 5.1 Participants and Apparatus

Eighteen adults (5 female) between 24 and 31 years old (average 27.2±2.1) participated in the study. All of them had normal sight or correct-to-normal sight and none of them reported himself/herself as color blind.

The apparatus was the same as that used in the Wizard-of-Oz study (Section 4.1.2), so was the eye-gaze tracking technology: we used an iPad Pro with true-depth camera; the eye-gaze tracking technology was implemented with the ARKit library, as previously described.

## 5.2 Design

We adopted a [$3 \times 2 \times 2$] within-participant design. The three independent variables were: (1) the target selection method with 3 levels (BayesGaze, CM, Dwell), (2) the target size with 2 levels (1 cm or $1.43°$, and 2 cm or $2.86°$), and (3) the $\alpha$ value of the Zipf's distribution with 2 levels ($\alpha = 1$, and $\alpha = 2$). The Zipf's distribution controls the distribution of the intended targets among the candidates.

For each selection method $\times$ target size $\times$ Zipf's law $\alpha$ combination, each participant performed 24 trials. When $\alpha = 1$, the frequencies of the 5 target candidates being the intended targets were 11, 5, 4, 3, 1; when $\alpha = 2$, these frequencies were 16, 4, 2, 1, 1. We included two $\alpha$ values to evaluate whether the skewness of the target distribution affects selection performance. Among a set of 24 trials, the distance between the target and the starting bar was either 4 cm or 5 cm with 50% probability for each distance, and the target was either above or below the starting bar, also with 50% probability for each option.

## 5.3 Procedure

For each trial, the participant was instructed to select one of the five adjacent horizontal bars displayed on the iPad screen via eye-gaze. The tracked gaze position was rendered as a cross-hair cursor on the display, as shown in Figure 5. The target to be selected was shown in blue and other targets in cyan. A starting bar was also displayed, which served as the starting position for the gaze input. Prior to starting a trial, the participant was asked to move the cursor into the starting bar which was initially displayed in gray. The bar turned to green after three seconds, signaling the start of a trial. The participant then moved the cursor to select the target bar on the screen. The selected target then turned dark. If the user selected the wrong target, or did not select any target after 5 seconds after the beginning of the trial, it was considered a miss. The participant moved to the next trial regardless of the outcome of the trial. To alleviate eye fatigue, the participant was allowed to take a break no longer than 2 minutes every 15 trials. Figure 5 shows a screenshot of the experiment and a participant performing a trial.

After each trial, BayesGaze updated the prior probability for each target candidate. We assumed that each condition corresponds to a particular interface, and when the experimental condition changes (e.g., target size, or $\alpha$ value in Zipf's distribution), we reset all the prior information.

The participants were guided to select the target as accurately and quickly as possible. At the end of the study, participants were asked to rate their preference over the three methods on a scale of 1 to 5 (1: dislike, 5: like very much). They also answered a subset of NASA-TLX [12] questions to measure the workload of the gaze target selection task, including about mental and physical demand. The rating of the workload was from 1 to 10, from least to most demanding. The experiment lasted about 50 minutes.

To counterbalance the independent variables, the methods were fully balanced based on all 6 possible orders. For half of the users, $\alpha$ was set to 1 for the first half of the trials, and to 2 for the other half. For the other half of the users, it was the opposite order. Other factors were randomized. In total, we collected 18 users $\times$ 3 methods $\times$ 2 target sizes $\times$ 2 $\alpha$ $\times$ 24 trials = 5184 trials.

## 5.4 Results

We evaluate the performance of the BayesGaze, CM, and Dwell by the success rate and selection time.

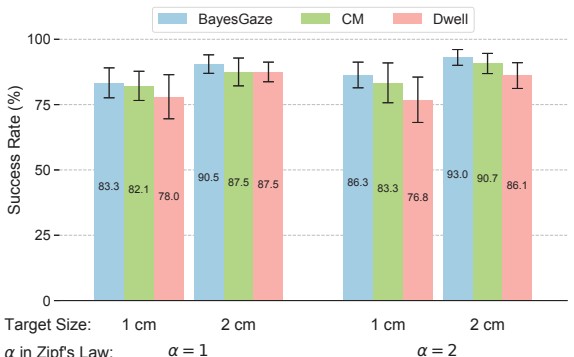

(a) Success rate by target size $\times$ Zipf's Law's $\alpha$

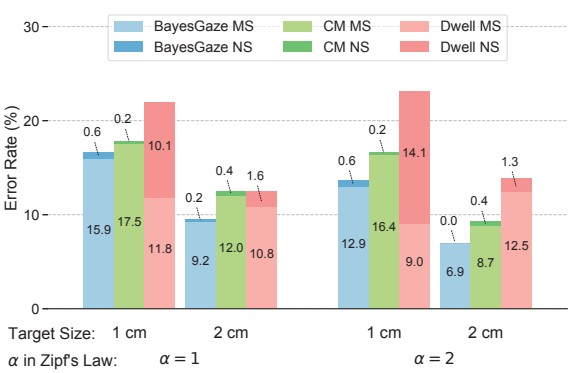

(b) Decomposition of the error rate for target size $\times$ Zipf's Law's $\alpha$

Figure 6: The average success rate with 95% CI and the decomposition of the error rate (Mis-Selection (MS) and Non-Selection (NS))

### 5.4.1 Success Rate

The success rate measures the ratio of correct selections over the total number of trials. The results (Figure 6a) show that: 1) BayesGaze always has the highest success rate and Dwell has the lowest success

rate, which confirms the effectiveness of Bayesian approach and the benefit of using the prior. 2) Large targets (2cm) have higher success rate than small targets (1cm), because it is much easier to move one's gaze into a large target.

A repeated measures ANOVA on success rate shows two significant main effects: target selection method ($F_{2,34} = 11.45, p < 0.001$) and target size ($F_{1,17} = 30.76, p < 0.001$). The test does not show a significant main effect of Zipf's Law's $\alpha$ ($F_{1,17} = 1.722, p = 0.207$). There is no significant interaction effect. Pairwise comparisons with Holm adjustment [13] on the success rate show significant differences between BayesGaze vs. Dwell ($p < 0.01$), CM vs. Dwell ($p < 0.05$), and BayesGaze vs. CM ($p < 0.05$).

The overall mean$\pm$95% confidence interval (CI) of success rate among all target sizes and $\alpha$ is 88.3%$\pm$3.6 for BayesGaze, 85.9%$\pm$4.3 for CM, and 82.1%$\pm$5.2 for Dwell. In total, BayesGaze improves the success rate by 6.2% over Dwell, and by 2.4% over CM.

In addition to the success rate, we also look into the error rate, which measures the ratio of the cases where the right target is not selected. There are two types of errors: (1) Mis-Selection (MS), where a wrong target is selected, and (2) Non-Selection (NS), where no target is selected. We examine the error rates of these two types of errors separately. Figure 6b shows the decomposition of the error rate. The major part of the error rate of BayesGaze and CM comes from mis-selection, and the same for Dwell when the target size is 2 cm. However, when the target size is 1 cm, Dwell suffers from not selecting any target. The result implies that using a Bayesian framework can alleviate the problem of not being able to select target.

With BayesGaze, a potential side effect of incorporating the prior might be that less frequent targets are more difficult to select. Table 2 shows the success rates by target frequency. Although the success rates for items with a frequency of 1 are lower than for the high frequency items, they are still near 80%. A repeated measures ANOVA does not show significant main effects of frequency on success rate for BayesGaze ($F_{9,153} = 0.776, p = 0.639$), CM ($F_{9,153} = 1.248, p = 0.27$), or Dwell ($F_{9,153} = 0.669, p = 0.736$), indicating that this potential side effect is minor.

### 5.4.2 Selection Time

Figure 7 shows the results for selection time, which measures the time to select the target from the start of the trial. As with the success rate, we observe that: 1) BayesGaze has the lowest selection time, and Dwell has the longest one; 2) Small targets (1cm) take longer to select than large ones (2cm), especially for Dwell.

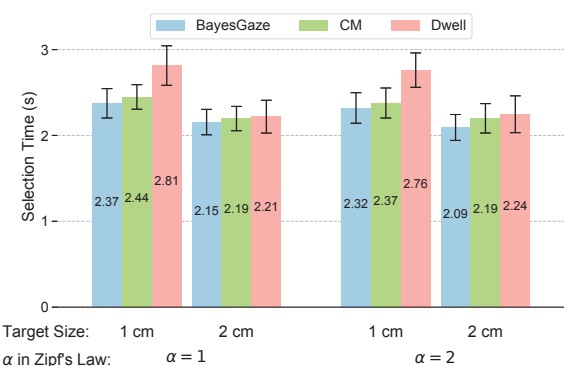

Figure 7: The average selection time (with 95% CI) by target size $\times$ Zipf's Law's $\alpha$

A repeated measures ANOVA on selection time shows two significant main effects: target selection method ($F_{2,34} = 21.19, p < 0.001$)

| Target Selection Method | Frequencies when $\alpha = 1$ | | | | | Frequencies when $\alpha = 2$ | | | | |
|---|---|---|---|---|---|---|---|---|---|---|
| | 11 | 5 | 4 | 3 | 1 | 16 | 4 | 2 | 1 | 1 |
| BayesGaze | 88.1 | 86.1 | 88.9 | 84.3 | 77.8 | 90.6 | 87.5 | 90.3 | 88.9 | 83.3 |
| CM | 85.6 | 82.8 | 84.0 | 86.1 | 86.1 | 85.9 | 87.5 | 93.1 | 88.9 | 88.9 |
| Dwell | 83.1 | 85.6 | **75.7** | 84.3 | 88.9 | 79.9 | 85.4 | 83.3 | 86.1 | 83.3 |

Table 2: The success rate (%) for different target selection frequencies (the lowest success rate is marked in bold)

and target size ($F_{1,17} = 116.9, p < 0.001$). The test does not show a significant main effect of Zipf's Law's $\alpha$ ($F_{1,17} = 1.685, p = 0.212$). The only significant interaction effect is target size × target selection method ($F_{2,34} = 31.81, p < 0.001$). Pairwise comparisons with Holm adjustment on selection time show significant differences for BayesGaze vs. Dwell ($p < 0.001$) and CM vs. Dwell ($p < 0.01$). The pairwise comparisions does not show a significant difference for BayesGaze vs. CM ($p = 0.09$).

The overall mean±95% CI selection time among all target sizes and $\sigma$ is 2.23±0.15 seconds for BayesGaze, 2.30±0.15 seconds for CM, and 2.49±0.18 seconds for Dwell. In total, BayesGaze can save 10.4% selection time over Dwell, and 3% over CM.

### 5.4.3 Subjective Feedback

The result of subjective feedback is shown in Figure 8. For overall preference, the median ratings for BayesGaze, CM and Dwell are 4, 3.5 and 3 respectively. BayesGaze has the highest median rating. For mental and physical demand, the medians are 6.5 and 5.5 for BayesGaze, 6 and 6 for CM, and 7.5 and 7.5 for Dwell. Nonparametric Friedman tests do not show significant main effects of selection method on three metrics: overall preference ($X_r^2(2) = 1.11, p = 0.57$), physical demand ($X_r^2(2) = 2.93, p = 0.085$), and mental demand ($X_r^2(2) = 5.24, p = 0.073$). The $p$ values for physical and mental demanding are approaching statistical significance.

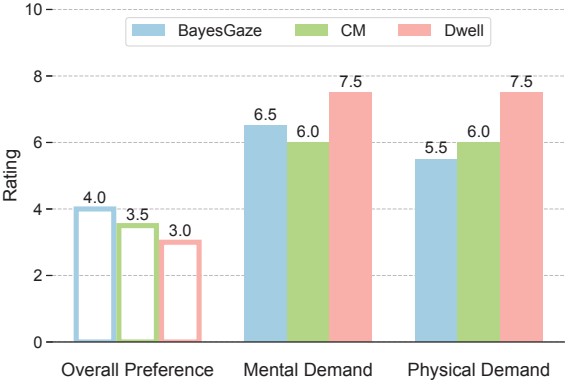

Figure 8: The median of subjective ratings of overall preference, mental demand and physical demand. For overall preference, higher ratings are better. For mental and physical demand, lower ratings are better.

### 5.5 Discussion

*Performance.* The experiment results show that BayesGaze outperformed both the Dwell and CM methods, in both selection accuracy and speed. BayesGaze improved the success rate of Dwell from 82.1% to 88.3%, i.e. a 6.2% increase, and reduced selection time from 2.49 seconds to 2.23 seconds,i.e. a 10.4% deduction. BayesGaze also improved the success rate of CM by 2.4%, and reduced the selection time by 3%. Pairwise comparisons with Holm

adjustment showed all these differences to be significant ($p < 0.05$), except for selection time between BayesGaze vs. CM ($p = 0.09$).

The promising performance of BayesGaze first shows that incorporating the prior significantly improves target selection performance. Compared with CM, which can be viewed as BayesGaze without prior, BayesGaze performed better in both accuracy and speed across all conditions. This suggests that incorporating the prior distribution of targets is effective in improving the performance of gaze-based target selection tasks. Second, both BayesGaze and CM outperformed Dwell, indicating that accumulating the interest, which is represented by the posterior in BayesGaze and by the likelihood in CM, is also effective for gaze-based target selection.

*Prior.* Incorporating the prior might make less frequent targets more difficult to select, even though we did not observe it in our experiment, as shown in Table 2. There are several ways to prevent this potential problem: (1) Set a lower bound for the target frequency so that no target will become hard to select. (2) In real-world applications, leverage user actions to address the problem. For example, if the previous selection is incorrect (back/cancel action is performed immediately), reduce the probability of the incorrect target. (3) Similar to what we do in this paper, use a small $\sigma$ for the likelihood model in order to decrease interference between neighboring targets.

*Target Dimension.* This paper considers 1D targets to show that Bayes' theorem can be adopted to improve the performance of gaze-based target selection. In real applications, there are many linear menus on computers and smartphones where our method can be directly applied. However, the underlying principle (Equation 2 - 5) is not tied to a specific type of target and can also be used for 2D target selection. The main difference between 1D and 2D target selection lies in the likelihood function (Equation 5). For 1D targets, we adopted a 1D Gaussian; for 2D targets it should be replaced by a 2D Gaussian distribution. For the 2D Gaussian likelihood function, we need to decide the variance in the $X$ and $Y$ direction and a covariance between the $X$ and $Y$ direction. We can obtain the the variance and covariance by using a grid search method based on collected gaze trajectories for selecting targets. We can also fit the 2D Gaussian model to the collected gaze trajectories to obtain the parameters. The rest of the method, including updating priors, accumulating weighted posterior, and using Pareto optimization to balance accuracy and selection time will remain the same.

*Midas-Touch Problem.* The Midas-Touch Problem describes unintentional eye-gaze target selection when the user is reading content. Our method can work with existing approaches to solve the Midas-Touch problem in gaze target selection, for example: (1) We can use methods like [5, 44] to infer whether a user is reading content on the UI or controlling their gaze to select a target. These methods will classify gaze positions into content reading phase and target selection phase. BayesGaze can discard the gaze positions in the content reading phase, and use only the gaze positions in the target selection phase to decide the target. (2) We can increase the threshold of accumulated posterior for selection to mitigate the Midas-Touch problem. Reading content on UI tends to take a shorter period of time than controlling gaze to select a target. Increasing the threshold could prevent falsely activating the selection, and the actual threshold should be set based on specific scenarios. This approach is also adopted by dwell-based methods (e.g., [28]) to

mitigate the Midas-Touch problem.

*Scalability*. BayesGaze uses a gaze position buffer to store the gaze trajectory and empties it after each selection action. Our study (Figure 7) shows that most selections happen within 3 seconds, which takes only a small amount of memory to store gaze data. In real-world applications, we may set a rolling-window with a size of 3 seconds to store gaze position. It can then scale up and handle long gaze-based input.

## 6 CONCLUSION

In this paper, we introduced BayesGaze, a Bayesian approach to determining the selected target given an eye-gaze trajectory. This approach views each sampling point in a gaze trajectory as a signal for selecting a target, uses Bayes' theorem to calculate the posterior probability of selecting a target given a sampling point, and accumulates the posterior probabilities weighted by the sampling interval over all sampling points to determine the selected target. The selection results are fed back to update the prior distribution of targets, which is modeled by a categorical distribution with a Dirichlet prior. Our controlled experiment showed that BayesGaze improves target selection accuracy from 82.1% to 88.3% and selection time from 2.49 seconds per selection to 2.23 seconds over the widely adopted dwell-based selection method. It also improves selection accuracy and selection time over the CM method [4] (85.9%, 2.3 seconds per selection), a high-performance gaze target selection algorithm. Overall, our research shows that both incorporating the prior and accumulating the posterior are effective in improving the performance of gaze-based target selection.

## ACKNOWLEDGMENTS

We thank the anonymous reviewers for their insightful comments. This work was supported by NSF awards 1815514, 1805076, 1936027, NIH awards R01EY030085, R01HD097188 and ALS Association grant 20-MALS-538.

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
