# OpenReview forum: "BayesGaze: A Bayesian Approach to Eye-Gaze Based Target Selection"
_graphicsinterface.org/Graphics_Interface/2021/Conference/Second_Cycle — GI 2021_

### Official Review · Reviewer_iyi1 · 2021-04-30
**This paper introduced BayesGaze, a Bayesian approach to determine the selected target given an eye-gaze trajectory.**

**Rating:** 7
**Confidence:** 3

**Review:**

Contribution:
This paper introduced BayesGaze, a Bayesian approach to determine the selected target given an eye-gaze trajectory. In a controlled experiment, they showed that BayesGaze improves target selection accuracy (from 82.1% to 88.3%) and speed (from 2.49 seconds per selection to 2.23 seconds) over a dwell-based selection method and the Center of Gravity Mapping (CM) method.

Review:
The authors tackled an interesting problem of selecting a target with the gaze by calculating the posterior probability of selecting a target given sampling points in the gazing trajectory. The motivation is clearly explained, and the paper is mostly well-written. The evaluation is moderately well-designed. It also covers related work in the area. I have some minor concerns about the paper (detailed below) but that can be addressed:

Concerning the presentation, a few things are unclear to me:

1) How did authors detect if the gaze trajectory accidentally leaves a target?

2) The statement “BayesGaze resumes the accumulation of selection interest from where it left” requires further explanation on how this is improving the performance of the selection. I believe it will increase the number of false alarms since it keeps on accumulating even if a user accidentally reaches the selection area.

Concerning the evaluation,

1) Did the authors observe a performance difference between different screen regions? If yes, were the differences significant? It would have been great to see this evaluation in the paper.

2) Also, the authors mentioned that their algorithm works for both 1D and 2D targets, did they evaluate this in an experiment?
I would like to see some discussion about the performance difference between seen/similar and unseen/new trajectory during the selection process.

3) What is the sampling interval and how did they select it? Did the authors perform an ablation study for the same?

4) Since the authors evaluated their approach only for the task where the target is a horizontal bar and gaze motion is vertical, how their algorithm will perform for vertical target and horizontal gaze motions. Some discussion about this could strengthen the paper.

Overall, I am positive about the paper as it explores an interesting approach to select a target with eye gazing. That said, I would argue for accepting it.

---

### Official Review · Reviewer_MJEp · 2021-05-04
**Reasonable technique and study, although narrowly focused**

**Rating:** 8
**Confidence:** 4

**Review:**

The paper contributes a new gaze-based pointing technique where the selection action is governed by a Bayes-theorem-based mechanism. A controlled study indicates that the new technique is more accurate and faster than a dwell-based technique or a previous Bayes technique.

Overall the paper is well written, the technique is clearly explained, and the study's methodology seems sound; on the whole I am in favour of acceptance. However, the following issues and limitations need to be addressed:

* The paper implicitly claims that selection based only on gaze is a requirement, but does not justify this claim; the paper does mention other mechanisms but does not consider them further and does not compare the Bayes technique to techniques such as blink-based or EMG-based or head movement or speech-based selection. The authors need to explain why it is reasonable to restrict their consideration to gaze-only selection (particularly given recent results such as the Sidenmark and Gellerson reference in the paper - which I note is not discussed or cited in the related work!)
- Mateo, J. C., San Agustin, J., & Hansen, J. P. (2008). Gaze beats mouse: hands-free selection by combining gaze and emg. In CHI'08 extended abstracts on Human factors in computing systems (pp. 3039-3044).

* In addition, more introduction to settings where gaze-based pointing is the only option would be valuable; the paper simply assumes that this is an open problem, but it takes a few minutes to think of reasonable scenarios where other modalities are not available. Since there are several broad types of these that have very different requirements (e.g., hands-occupied interaction vs. motor impairments) the authors should make clear what scenario they are designing for.

* The paper only studies 1D interaction, and there is little discussion given as to how the technique will generalize to 2D; the authors state that "the underlying principle works in 2D" but do not provide adequate evidence or discussion. Given that the vast majority of targets in interactive systems involve 2D pointing, this is a substantial limitation, and it is not sufficient to simply assume that the technique will generalize correctly.

* It is not clear why the authors use the term "Wizard of Oz" when describing the first phase of their study. This is not a Wizard of Oz methodology (unless something substantial is missing from the description). https://en.wikipedia.org/wiki/Wizard_of_Oz_experiment

Overall, the submission is solid but does not adequately justify the narrowness of the investigation (both in terms of pure gaze selection, and in terms of 1D targeting). The former of these, at least, should be fixable in the revision cycle, and it may be possible to better justify either the motivation for focusing only on 1D or the generalization to 2D.

---

### Official Review · Reviewer_PrSC · 2021-05-04
**Probable reject**

**Rating:** 4
**Confidence:** 4

**Review:**

While this submission presents an internally valid research effort I have major doubts on how the work scales to real world usability scenarios, and thus on how valid and interesting the findings will be to the GI audience and readership.

The paper's main strength is the introduction of a Bayesian approach to gaze target selection. There is a body of past work on Bayesian frameworks for eye tracking and eye control but, as far as I know, not in the HCI context of target selection on a phone/tablet.

The paper has a lot going for it: it is well written, the related work section is thorough, and the researchers seem to follow an overall solid methodology (though I was quite confused by the WOz  approach to study 1, please see more below).

The main reason I am not supportive of acceptance is what I believe to be a lack of attention to external validity in the current manuscript. Several aspects of the work seem to be narrowing down the scope of the research to lab settings with little insight on whether the findings will have much meaning in the real world, and with little attention to how the findings might scale to real task scenarios and real settings. This is a major flaw in my eyes and I hope the authors will at least attempt to provide more insight and discuss these aspects and possible implications of their work in case the paper is accepted to GI'21.

More specifically, the 1D target selection tasks, which are core to the paper and its two studies, are very limiting and questionable. I was not sure if the authors suggest that a 1D target selection task is valid in some settings? And if yes, which ones? If the 1D target selection task is just a precursor to 2D target selection, why are the authors convinced that their findings will scale?

While the paper's motivating Figure 1 seems to suggest a phone form factor, the study experiments are using an ipad which can maybe qualify as a large screen phablet, placed on a desktop stand. Again, the experimental settings are not invalid, but are very constrained, and arguably do not map to real world phone usage settings. I was hoping the authors could provide more insight on how their work could possibly scale to actual usage scenarios, where a a much smaller handheld phone is being handled and addressing various head poses (and yes, mostly 2D target selection tasks...) and maybe thinking of hybrid approaches that do not rely only on gaze for target selection. To clarify, I am not necessary suggesting an expansion of the studies, rather a much richer discussion of the limited scope studies presented in the paper in a wider context, allowing the reader to think of ways of applying the paper's findings in real world settings.

I was confused by the authors suggesting that they used a Wizard of Oz approach for their first study (only?). The details are lacking and I am uncertain how a wizard could operate such a study, and whether the introduction of a wizard would not compound the findings. In case of acceptance I suggest the authors please elaborate on these points.

Finally, a video figure would have helped clarify aspects of the studies and the findings.

---

### Meta-Review · Area_Chair_oBGh · 2021-05-06

**Recommendation:** Accept
**Confidence:** 4

**Metareview:**

The paper clearly has positive and negative aspects as identified by the external reviews. On the positive side, the paper is well written and thorough, the study has been carried out well, and the technique solves a recognized problem in the area. On the negative side, all reviewers noted that the limitations of the research are also substantial - particularly the relatively narrow focus of the work as well as its external validity in terms of the device chosen, the use of 1D rather than 2D targeting, the lack of comparison to other ways of providing a selection action, and specifics of the technique's effectiveness for different types and locations of targets.

On balance, reviewers lean slightly towards acceptance, but with very strong caveats about issues that would need to be addressed in any revision of the paper. The individual reviews clearly indicate what needs to be done to improve the scoping of the paper and further discussion of limitations and external validity.

---

### Decision · Program_Chairs · 2021-05-08

Accept